# Beeswax Alcohol (BWA, Raydel^®^) Improved Blood Oxidative Variables and Ameliorated Severe Damage of Zebrafish Kidneys, Testes, and Ovaries Impaired by 24-Week Consumption of a High-Cholesterol and High-Galactose Diet: A Comparative Analysis with Coenzyme Q_10_

**DOI:** 10.3390/ph18010017

**Published:** 2024-12-26

**Authors:** Kyung-Hyun Cho, Ashutosh Bahuguna, Ji-Eun Kim, Yunki Lee, Sang Hyuk Lee, Cheolmin Jeon, Cheol-Hee Kim

**Affiliations:** 1Raydel Research Institute, Medical Innovation Complex, Daegu 41061, Republic of Korea; 2Department of Biology, Chungnam National University, Daejeon 34134, Republic of Korea

**Keywords:** beeswax alcohol (BWA), coenzyme Q_10_ (CoQ_10_), oxidative variables, senescence, oocytes, cholesterol, kidney, ovary, testis

## Abstract

Objectives: The present study describes the comparative effect of 24-week supplementation of beeswax alcohol (BWA, Raydel^®^, 0.5% and 1.0%, wt/wt) and coenzyme Q_10_ (CoQ_10_, 0.5% and 1.0%, wt/wt) on plasma oxidative variables and the prevention of organ injury in adult zebrafish subjected to a high-cholesterol (HC, 4%, wt/wt) and -D-galactose (Gal, 30%, wt/wt) diet. Methods: Adult zebrafish were fed various HC+Gal diets enriched with either BWA or CoQ_10_. After 24 weeks of dietary intervention, blood and organs were harvested for subsequent biochemical and histological evaluations. Results: The HC+Gal-elevated plasma oxidative variables were reverted by the consumption of BWA, marked by the lowest plasma malondialdehyde (MDA) level and highest sulfhydryl content. The HC+Gal-impaired zebrafish swimming ability (staggering movement) was substantially recovered by BWA, manifested by a ~three-fold (*p* < 0.001) enhancement in swimming distance and speed. Also, the intake of BWA affected the morphology of HC+Gal-compromised kidney and induced histological changes by mitigating reactive oxygen species (ROS) production and cellular senescence, which was markedly more effective than the results seen in the CoQ_10_ group. Likewise, BWA proved effective in preventing HC+Gal-induced testis damage, apparent in the 48.3% (*p* < 0.05) higher spermatozoa and 26.3% (*p* < 0.01) reduced interstitial space between the seminiferous tubules. BWA substantially prevented HC+Gal-induced ovary damage by suppressing oxidative stress, lipid deposition and senescence, leading to the restoration of mature vitellogenic oocyte counts. Conclusion: BWA demonstrated a greater ability than CoQ_10_ to enhance plasma antioxidant status, suppress ROS generation, delay organ aging and alleviate HC+Gal-induced adversity in zebrafish.

## 1. Introduction

Many investigations have established that extensive consumption of D-galactose (Gal) is associated with severe damage in major organs such as the brain, liver, kidneys, ovaries, and testes via the formation of galactitol, which causes oxidative stress and mitochondrial dysfunction [1,2]. It is well known that the accumulation of galactitol leads to depletion of reduction power to defend against oxidative stress, consequently exacerbating neurotoxicity, hepatotoxicity, and cell aging [3,4]. Due to the build-up of reactive oxygen species (ROS)/free radicals and the hypertonic nature of galactitol, severe oxidative damage occurs in various cell membranes, proteins, and mitochondrial DNA [5,6]. In particular, the long-term consumption of D-galactose induces significant oxidative stress and the formation of advanced glycation end products (AGEs), which contribute to brain aging in animal models, closely resembling the aging process observed in the human brain and liver [7,8]. Besides the effect on the brain and liver, it was documented that D-galactose causes the aging of testicular tissue, the thymus, and the spleen, with severe oxidative damage in a murine model [3,9,10].

Beeswax alcohol (BWA) is a blend of long-chain aliphatic alcohol (LCAA) extracted from honey beeswax [11]. BWA is widely recognized for its potent antioxidants and anti-inflammatory properties, which enable it to exert hepatic and gastroprotective roles and prevent the oxidation of protein and lipids in the cell membrane [12,13,14]. Moreover, in vitro experiments highlighted BWA’s role in inhibiting the oxidation of low-density lipoprotein (LDL) [15] while protecting the activity of high-density lipoprotein (HDL)-associated paraoxonase (PON) and reducing HDL glycation [16]. The administration of BWA intake demonstrated therapeutic potential against dyslipidemia by lowering the total cholesterol (TC) and triglyceride (TG) levels while increasing the HDL-C levels in hyperlipidemic zebrafish exposed to both mild [11] and high doses of D-galactose [17].

Like BWA, coenzyme Q_10_ (CoQ_10_) is a hydrophobic substance known for its various activities, such as its anti-inflammatory, antioxidant, and hepatoprotective properties [18,19]. In addition, CoQ_10_ was reported to significantly contribute to the maintenance of the plasma lipoprotein profile in a study on dyslipidemic Chinese adults [20]. Despite the several benefits of BWA and CoQ_10_, limited studies are available to precisely describe their comparative effect against high cholesterol- and high galactose-guided events. However, recently, we reported a substantially better effect of BWA over CoQ_10_ in protecting the brain and liver against high cholesterol and galactose (HC+Gal)-triggered oxidative stress, inflammation, and cellular senescence in adult zebrafish [17]. A potent protective effect was noticed in the brain, where BWA reduced ROS production, apoptosis, and senescence; protected the myelin sheath integrity; and restored cognitive changes in the zebrafish impaired by long-term dietary supplementation with HC+Gal [17]. In continuation of the previous study, the current study assessed the comparative pharmacological effect of BWA and CoQ_10_ on the plasma oxidative parameters and organs like the kidney, testis, and ovary of zebrafish consuming a high-cholesterol (HC, 4% wt/wt) and high-galactose (Gal, 30% wt/wt) diet. The zebrafish was chosen as the model species in light of its higher genetic similarity with higher vertebrates [21], which has been widely accepted as a suitable model for studying oxidative damage in the brain, liver, kidney, and reproduction systems and has a very similar lipid and lipoprotein metabolism to humans [22].

## 2. Results

### 2.1. Change in Motion Pattern and Swimming Behavior in Top-Down View Assay

The bird’s-eye-view motion assay (Figure 1) revealed that zebrafish in the ND group showed the highest swimming distance, around 5.3 m, and the most comprehensive swimming pattern around the rim of the wall (of the round tank), while the zebrafish in the HC alone group showed a decreased swimming distance (~2.7m) with a narrow swimming pattern along the half-rim wall of the tank. Furthermore, the HC+Gal group showed the shortest swimming distance (1.5 m), which was highly restricted to the tank’s center without approaching the tank’s wall, suggesting that galactose consumption along with the HC diet causes severe loss of swimming ability and sense of direction. By contrast, the co-consumption of 0.5% BWA and 1.0% BWA culminated in improved swimming distances of 4.7 m and 5.1 m, respectively, with a broader swimming pattern along the wall of the round tank. Precisely, the zebrafish in the 1.0% BWA group displayed swimming movements along the complete rim of the tank. The co-consumption of 0.5% and 1.0% CoQ_10_ also resulted in improved swimming distances of ~3.7 m and 3.3 m, respectively, which was remarkably ~2-fold more than the swimming distance noted in the HC+Gal group. Also, a much-improved swimming pattern was noticed in the CoQ_10_ group compared to the HC+Gal group; however, the swimming motion was almost restricted to half the rim of the tank. Compared to CoQ_10_, zebrafish in the BWA group, specifically in the 1.0% group, showed free movement across the tank with a significantly ~1.5-fold (*p* < 0.001) higher swimming distance than that of the 0.5% and 1.0% CoQ_10_ groups, respectively.

Alongside swimming distance, the highest swimming speed (8.8 cm/s) was noticed in the ND group, which was significantly reduced to 4.6 cm/s in the HC group and further reduced to 2.4 cm/s in the HC+Gal group. In contrast, the swimming speed was substantially restored in the CoQ_10_ and BWA consumption groups. The zebrafish in the 0.5% and 1.0% BWA groups displayed a swimming speed of 7.9 and 8.5 cm/s, respectively, which was remarkably ~3-fold better than the swimming speed observed in the HC+Gal group. Similarly, the intake of CoQ_10_ resulted in a noticeable increase in swimming speed, reaching 6.1 cm/s and 5.6 cm/s in the 0.5% CoQ_10_ and 1.0% CoQ_10_ groups, respectively. Nevertheless, BWA demonstrated a more substantial impact on swimming speed, as reflected by the ~39% higher swimming speed in the 1.0% BWA group than that of the 1.0% CoQ_10_ group.

### 2.2. Oxidative Extent and Antioxidant Status of Plasma

The thiobarbituric acid reactive substance (TBARS) assay revealed the highest malondialdehyde (MDA) level in the HC+Gal group (11.7 μM), which was significantly 1.7-fold and 1.5-fold more compared to the MDA level quantified in the ND (4.0 μM) and HC (7.8 μM) consumption groups (Figure 2A). The consumption of BWA displayed a dose-dependent effect to reduce the HC+Gal-elevated MDA level, as evidenced by the 6.3 μM and 5.5 μM MDA levels in the 0.5% and 1.0% BWA groups, which was significantly 1.9-fold (*p* < 0.01) and 2.1-fold (*p* < 0.01) lower compared to the MDA level detected in the HC+Gal group. Similarly, a positive effect of CoQ_10_ supplementation, a reduction in the plasma MDA level was observed, which was ~1.4-fold (*p* < 0.05) lower compared to the MDA level in the HC+Gal group, while compared to 1.0% CoQ_10_, the 1.0% BWA group displayed a ~33.2% (*p* < 0.001) reduced MDA level.

The plasma sulfhydryl content was significantly reduced by 39.5% in the HC+Gal group compared to that in the ND group (Figure 2B). The HC+Gal-diminished sulfhydryl content was restored in the BWA consumption group, evidenced by the significantly, ~32% (*p* < 0.01), elevated sulfhydryl content in the 0.5% and 1.0% BWA groups. In contrast, CoQ_10_ consumption displayed a non-significant effect on the plasma sulfhydryl content, disturbed by the consumption of HC+Gal. The combined results of the plasma analysis attested to BWA’s greater efficacy compared to CoQ_10_ in diminishing HC+Gal oxidative stress markers.

### 2.3. Change in Organ Weight: Kidneys, Testes, and Ovaries

As shown in Figure 3B,E, the HC alone and HC+Gal groups displayed 1.4-times and 1.7-times augmented kidney weight compared to the ND alone group, suggesting that the HC+Gal diet exacerbates the kidney burden more than the HC alone diet. The BWA 0.5% and 1.0% groups displayed 15% and 27% less kidney weight relative to the HC+Gal group, suggesting that supplementation with BWA alleviated enlargement of the kidneys and damage caused by the HC+Gal diet. Like the BWA group, the 1.0% CoQ_10_ group displayed a substantially, 29% (*p* < 0.001), reduced kidney weight compared to the HC+Gal group; however, 0.5% CoQ_10_ consumption displayed a non-significant effect.

As shown in Figure 3C,F, a non-significant change in the testes size and weight was observed between the groups compared to the HC+Gal group. Similar to the findings of the testis, no substantial change in the ovary weight was detected in different groups (Figure 3D,G). Nonetheless, the morphological analysis showed a slightly reduced ovary size in the HC+Gal consumed group compared to the other groups (Figure 3D).

### 2.4. Histological Examination and Senescence Assay in Kidney

As depicted in Figure 4A, a disorganized tubular structure was observed in the HC- group that substantially elevated with the co-consumption of Gal (HC+Gal). The HC+Gal group exhibited an enlarged tubular lumen (indicated by a blue arrowhead) at multiple locations, along with frequent degeneration and debris in the lumen. The BWA, mainly at 1.0%, significantly restored the HC+Gal-induced kidney impairment reflected by the well-organized distal and tubular structure; however, elevated space in the tubular lumen was noticed at some places. In contrast, the CoQ_10_-supplemented groups showed limited preventive effect, where the occasional presence of the lumen debris and the dilated tubular lumen was noticed.

DHE staining suggested abundant ROS generation in the HC+Gal group, which was significantly 10.4-fold more than the ROS level quantified in the ND group (Figure 4B,E). The supplementation of 0.5% and 1.0% BWA effectively reduced the HC+Gal-elevated ROS level, as reflected by the 3.2-fold (*p* < 0.001) and 10-fold (*p* < 0.001) diminished DHE fluorescent intensities, respectively, compared to that of the HC+Gal group. Like BWA, the 0.5% CoQ_10_ consumption group showed a substantial effect against HC+Gal-induced ROS generation. Surprisingly, 1.0% CoQ_10_ supplementation failed to prevent HC+Gal-induced ROS production.

AO staining revealed that the HC+Gal group exhibited the highest level of apoptosis, indicated by a 14-fold higher AO-stained area than the ND group (Figure 4C,E). Supplementation of BWA at both 0.5% and 1.0% concentrations efficiently counters HC+Gal-provoked apoptosis, evidenced by the 2.9-fold and 9.2-fold lower AO-stained areas in the 0.5% and 1.0% BWA groups, compared to those in the HC+Gal group (Figure 4C,E). Unlike BWA, the consumption of CoQ_10_ failed to prevent HC+Gal-induced apoptosis.

The SA-β-gal-stained area was significantly increased by 3.8-fold (*p* < 0.001) and 1.5-fold (*p* < 0.001) in the HC+Gal group compared to those in the ND and HC groups, respectively, indicating that HC+Gal consumption promotes cellular senescence in the kidneys (Figure 4D,F). Supplementation of BWA and CoQ_10_ at both doses (0.5% and 1.0%) adequately prevented HC+Gal-provoked cellular senescence. In the BWA, 0.5%, and 1.0% groups, notably 1.6-fold and 2.3-fold lower cellular senescence was quantified than in the 1.0% CoQ_10_ group, attesting to the substantially better impact of BWA over CoQ_10_ in restraining HC+Gal-triggered cellular senescence.

### 2.5. Histological Analysis and Senescence Assay in Testis Tissue

As depicted in Figure 5, the ND group displayed normal testis histology with an adequately arranged tubular structure that was substantially compromised by the consumption of HC and HC+Gal diets, as evident by the ~1.4-times (*p* < 0.01) elevated interstitial space between the seminiferous tubules and the ~1.5-times diminished spermatozoa area compared to the ND group (Figure 5A,B). The supplementation of BWA, mainly at 1.0%, significantly restricts HC+Gal-induced testis damage, as reflected by the 26.1%-reduced interstitial space between the seminiferous tubules and 48.3% (*p* < 0.05) higher spermatozoa than those in the HC+Gal group (Figure 5F,G). In contrast, CoQ_10_ consumption at 0.5% and 1.0% showed no significant therapeutic impact on the testis damage induced by HC+Gal.

DHE and AO staining revealed the heightened production of ROS and apoptosis in response to the consumption of HC+Gal, which was significantly 1.8-fold and 3.8-fold higher than the ROS and apoptosis detected in the ND group (Figure 5C,D,H). The supplementation of BWA and CoQ_10_ showed a substantial preventive role against the HC+Gal-triggered ROS generation and apoptosis in the testis. Notably, a substantially 1.4-fold and 2.0-fold lower DHE and AO-stained area was quantified in the 1.0% BWA group compared to the 1.0% CoQ_10_ group.

SA-β-gal staining (Figure 5E,I) suggests a 15.5-fold and 1.5-fold higher prevalence of the senescent-positive cells in the HC+Gal group compared to in the 1.0% BWA and 1.0% CoQ_10_ groups, testifying to the protective role of BWA and CoQ_10_. However, the BWA effect is much superior to that of CoQ_10_ to minimize the HC+Gal-triggered senescence, as reflected by an 8.4-fold and 10.5-fold reduced senescent-stained area in the 1.0% BWA consumption group compared to the 0.5% and 1.0% BWA groups, respectively.

### 2.6. Histological Analysis of Ovaries

The histological analysis of the ovaries revealed a significant detrimental effect of HC+Gal on the ovary cellular structure by elevating the pre-vitellogenic oocyte counts by 1.2-fold and reducing the early-vitellogenic oocyte counts by 3.4-fold those in the ND group (Figure 6A,F,G). BWA consumption prevents HC+Gal-triggered ovary damage, as reflected by the ~1.2-fold reduced pre-vitellogenic oocyte counts in response to 0.5% and 1.0% BWA compared to those in the HC+Gal group. Likewise, early and mature oocyte counts were ~2.0-fold and ~3.6-fold higher in the BWA groups compared to those in the HC+Gal group. In contrast, CoQ_10_ failed to demonstrate any restorative effect on the oocyte counts impacted by the HC+Gal consumption of HC+Gal.

The DHE staining showed the effect of BWA and CoQ_10_ in curtailing HC+Gal-triggered ROS generation. A noticeable ~1.5-times diminished ROS level was noticed in the 0.5% and 1.0% BWA groups compared to the HC+Gal group (Figure 6B,H). Also, CoQ_10_ (0.5% and 1.0%) substantially diminished HC+Gal-triggered ROS production; however, the ROS levels remained approximately 20% higher (*p* < 0.001) in these groups compared to those in the 1.0% BWA group.

The highest apoptosis indicated by AO staining was observed in the HC+Gal group, which was remarkably 1.6-fold and 1.9-fold reduced in response to the consumption of 0.5% and 1.0% BWA (Figure 6C,H). Likewise, CoQ_10_ (0.5% and 1.0%) reduced HC+Gal-triggered apoptosis, although their efficacy was ~1.4 fold lower compared to the effect exerted by 1.0% BWA.

The SA-β-gal staining results demonstrated that BWA (at concentrations of 0.5% and 1.0%) significantly reduced HC+Gal-induced senescence by ~2.0-fold (*p* < 0.01) (Figure 6D,I). In contrast, CoQ_10_ did not exhibit a significant effect in preventing HC+Gal-induced cellular senescence.

Additionally, BWA significantly reduced HC+Gal-provoked lipid accumulation, evidenced by 1.3-fold and 2.4-fold diminished ORO-stained areas in the 0.5% and 1.0% BWA-supplemented group, compared to the HC+Gal-group (Figure 6E,I). In contrast, CoQ_10_ exhibited no effect against HC+Gal-induced lipid accumulation.

## 3. Discussion

It has been well known that high cholesterol consumption has been associated with several detrimental effects [23,24,25] that enhance oxidative stress and alter antioxidant status [26]. Likewise, high sugar consumption has been reported to cause oxidative stress [27], inflammation [28], and adverse events that substantially affect different organs [27]. Furthermore, the combined intake of sugar and HC results in greater exacerbation than either substance alone, making it an ideal dietary model for inducing obesity and other disorders that mimic the pathophysiology and progression of the disease in humans [29].

Galactose displayed severe toxic effects in adult zebrafish [11,17], leading to altered swimming behavior marked by retard swimming speed and paths of movement that concentrated in a fixed space. The consumption of BWA and CoQ_10_ showed a substantial preventive impact on the recovery of HC+Gal-altered swimming behavior; however, a higher effect of BWA than CoQ_10_ on HC+Gal-inducted adversity was noticed. It has been well recognized that galactose-induced ROS generation [30] and impaired cellular antioxidants [31] are the key contributors to brain aging and cognitive changes [30,32,33]. Substances that have a positive impact on the brain’s oxidative parameters and antioxidants, such as SOD, CAT, and glutathione peroxidase (GSH-px), lead to the recovery of brain damage and cognitive changes provoked by the galactose [31]. We believe that BWA has a better effect in restoring HC+Gal-hampered swimming behavior than CoQ_10_ due to its higher antioxidant impact [15]. Also, a direct positive modulatory effect of BWA on cellular antioxidants like superoxide dismutase (SOD), catalase (CAT), and glutathione peroxidase (GSH-px) has been documented [14,15,34] and strengthens the present finding that, owing to substantial antioxidant activity, BWA protects zebrafish against the HC+Gal-induced challenges. Besides the induction of oxidative stress, galactose consumption has been noticed to cause glycation, leading to the generation and accumulation of AGEs in the brain [30,35] and, consequently, brain damage and cognitive changes [36,37]. Therefore, a substance that can prevent protein glycation will have a definite effect on preventing glycation-induced detrimental effects. The accumulating literature suggests the inhibitory effect of BWA in preventing protein glycation, while CoQ_10_ was found ineffective [15]. Moreover, compared to CoQ_10_, BWA displayed a substantial protective effect against carboxymethyllysine (CML, a typical AGE)-induced toxicity in zebrafish embryos [15]. These findings support the current results, suggesting that BWA, due to its antiglycation properties and significantly more vigorous antioxidant activity than CoQ_10_, mitigates HC+Gal-induced swimming alterations in zebrafish. 

The plasma analysis revealed substantially higher MDA levels in the HC+Gal group, along with a reduced plasma sulfhydryl content, underscoring the impact of HC+Gal on the induction of oxidative stress. The consumption of CoQ_10_ and BWA effectively mitigated the HC+Gal-induced MDA level. Similarly, plasma sulfhydryl is a crucial oxidative stress marker [38,39], and its diminished levels have been associated with different pathological conditions [40]. Following HC+Gal supplementation, a reduction in plasma sulfhydryl content was noticed, which was significantly elevated by the intake of BWA; however, CoQ_10_ consumption did not restore the HC+Gal-disrupted plasma sulfhydryl content. The collective results of the plasma analysis depict a low level of MDA and higher sulfhydryl content in BWA than the CoQ_10_ directed the higher antioxidant nature of BWA over CoQ_10_ that effectively prevents the HC+Gal induced oxidative stress. These results align with those of earlier studies documenting the impact of BWA on the plasma oxidative variables and its modulatory effect on cellular antioxidants [41,42].

Severe damage to the kidney, testis, and ovary have been noticed in response to HC+Gal consumption, which is consistent with the literature documenting the adverse effects of HC and Gal in different organs [11]. Consistent with the observations of the plasma analysis, BWA consumption showed a significant curative impact in preventing the HC+Gal-induced impairment of kidneys and reproductive organs. Notably, in all organs, a high generation of ROS was noticed in response to HC+Gal consumption, which has been recognized as a key culprit for the induction of apoptosis [43,44] and senescence [45,46]. It is apparent from the current study that the higher antioxidant efficacy of BWA counters the HC+Gal-provoked ROS generation, which leads to the inhibition of apoptosis and senescence and the prevention of organ damage. The results are supported by earlier reports documenting the protective effect of antioxidants against Gal-induced kidney damage by minimizing oxidative stress [47]. In addition, the higher plasma sulfhydryl content in the BWA group suggests better kidney health, as higher sulfhydryl content has been associated with a decreased risk of kidney disease [38]. A visible impact of high cholesterol and galactose has been described for testis damage by impairment of the testicular structures and interstitial area between the seminiferous tubules, altered spermatogenesis [48], and decreased sperm count [49]. Also, galactose consumption has been documented to elevate oxidative stress and the inhibition of cellular antioxidant activities such as those of SOD and GSH-px, leading to severe damage to the testis [48]. BWA, owing to its antioxidant nature, suppresses the HC+Gal-posed oxidative stress, preventing testicular damage. The results aligned with a previous study highlighting caffeic acid antioxidant properties, which countered Gal-induced oxidative stress and safeguarded the testes [48]. Similarly, oxidative stress impacts the physiology of the ovary and oocyte quality and causes female reproductive health disorders [50], while the substance reduces oxidative stress and improves antioxidant status to prevent ovary damage [50]. BWA, due to the inhibitory effect of ROS generation and its inductive effect on cellular antioxidants, improves the oxidative stress environment in different organs, protecting against HC+Gal-induced damage, alongside its established role in preventing gastrointestinal and joint health of humans [51]. The current findings decode BWA’s efficacy in protecting vital organs and offer promising possibilities for the consumption of human and veterinary supplements to manage health issues linked to high cholesterol and sugar consumption. Further research and trials are essential to confirm its efficacy across different veterinary species.

## 4. Materials and Methods

### 4.1. Materials

Beeswax alcohol (BWA) (Cat#330020123) was extracted from the block beeswax of *Apis mellifera* (mainly mellifera lineage) and was complimentarily provided by Raydel^®^ Australia Pty. Ltd. (Sydney, NSW, Australia). Detailed specification of the BWA is provided in Appendix A. All the other chemicals and reagents were of analytical grade. The list of chemicals used is listed in Supplementary Method S1.

### 4.2. Zebrafish Husbandry and Production of Embryos

Adult ~16-week-old zebrafish were cultured in an aerated water tank equipped with a circulated water supply. The water temperature was maintained at 28 ± 1 °C, and a 14 h light/10 h dark photoperiod was used. The animal experiments were performed according to the suggested guidelines of the Animal Care Committee and approved by the Raydel Research Institute (RRI, approval code RRI-23-007, approval date 27 July 2023). Zebrafish were fed the normal diet (ND, tetrabit, Gmbh D4930, Melle, Germany).

### 4.3. Different Diets’ Preparation and Consumption by Adult Zebrafish

The normal diet (ND) of tetrabits was purchased from Tetrabit Gmbh (Melle, Germany). The ND was mixed with cholesterol (4%, wt/wt) to make the high-cholesterol (HC) diet following a previously described method [11]. The HC diet was blended with galactose (Gal, 30%, wt/wt) to form an HC diet infused with galactose (HC+Gal) [11,17]. The HC+Gal diet was blended with BWA (0.5% or 1.0% wt/wt) and CoQ_10_ (0.5% or 1.0% wt/wt) to make the four different HC+Gal diets supplemented with BWA and CoQ_10_.

Zebrafish (16 weeks aged, n = 392) were allocated to the different cohorts, named Group I (n = 56, ND), Group II (n = 56, HC), Group III (n = 56, HC + Gal + 0.5% BWA), Group IV (n = 56, HC + Gal + 1.0% BWA), Group V (n = 56, HC + Gal + 0.5% CoQ_10_), and Group VI (n = 56, HC + Gal + 1.0% CoQ_10_). The zebrafish in Group I were fed ND, while the zebrafish in Group II were fed the HC diet. Zebrafish in Groups III and IV were fed HC + Gal + 0.5% BWA and HC + Gal + 1.0% BWA, respectively. Similarly, the zebrafish in Groups V and VI were fed HC + Gal + 0.5% CoQ_10_ and HC + Gal + 1.0% CoQ_10_, respectively. Zebrafish in all groups (I–VI) were maintained on their respective diets for 24 weeks. Zebrafish in Groups II-VI were pre-fed with an HC diet for one month (before the experimentation) to induce hyperlipidemia. Figure 7 illustrates the experimental design for feeding across the groups.

### 4.4. Swimming Behavior Analysis

The swimming behavior of zebrafish following 24-week consumption of different diets was performed in a white circular water tank with a 35 cm diameter and a 11 cm height. The water tank was ~2/3 filled with water, and zebrafish from the different groups were allowed to swim in it. Before the swimming analysis, zebrafish were kept in the tank for 60 min to acclimate to the conditions. Afterward, zebrafish swimming activity was observed (1 min), and their swimming patterns were recorded using a digital camera (Canon EOS 90D, EFS 17-55 mm 1:2.8 IS-USM, ultrasonic lens, Tokyo, Japan) placed 100 cm above the water tank [as depicted in Appendix A. The recorded swimming activity was processed to analyze the path of motion, swimming distance, and speed using software (Any-Maze version 7.0, Kim & friends, Seoul, Republic of Korea).

### 4.5. Harvesting Blood and Tissues

Following 24 weeks of dietary intervention, zebrafish were euthanized via hypothermic shock [52]. Blood samples were promptly collected thereafter, and the organs (kidneys, testes, and ovaries) from the different groups were dissected and preserved separately in a 10% formalin solution for further analysis.

### 4.6. Estimation of Plasma Monoaldehyde (MDA) and Sulfhydryl Groups

The plasma MDA level was measured using the thiobarbituric acid reactive substance (TBARS) assay, following previously described methods with slight modifications [53]. Briefly, 50 μL of plasma (0.1 mg of equivalent protein) was mixed with 50 μL of trichloro acetic acid (TCA) and 100 μL of TBA. The mixture was subjected to 10 min at 95 °C, followed by an absorbance measurement at 560 nm.

The plasma sulfhydryl content was measured spectrophotometrically, following a previously established method with minor modifications [54]. In brief, 50 μL of plasma (0.1 mg of equivalent protein) was blended with 50 μL of 5,5-dithiol-bis-(2-nitrobenzoic acid) (DTNB) and incubated at room temperature. After 12 h of incubation, the absorbance at 412 nm was determined, and the sulfhydryl content was quantified using the DTNB molar extinction coefficient (ε) 1.36 × 10^4^ M^−1^ cm^−1^.

### 4.7. Histological Evaluation

The tissue sections (kidneys, testes, and ovaries) were sectioned (7 μm thick) using cryo-sectioning (Leica CM 1510S, Nussloch, Germany). The histological changes in the tissue section were investigated by hematoxylin and eosin (H&E) staining employing a previously outlined method [55]. Lipid accumulation in the tissue was determined by oil red O (ORO) staining [17]. Briefly, a 7 μm thick tissue section was submerged with the ORO stain followed by 5 min of heating at 60 °C and subsequent washing with 60% isopropanol and counter-stained hematoxylin (30 s). After washing with water, the section was observed under a microscope.

### 4.8. Staining for the Detection of Cellular Senescence, Reactive Oxygen Species, and Apoptosis

Cellular senescence in the tissue sections was assessed using the senescent-associated β-galactosidase (SA-β-gal) assay [17]. The tissue section, 7 μm thick, was coated with 250 μL of 5-bromo-4-chloro-3-indoly-β-D-galactopyranoside (X-gal, 5 mg/mL) and incubated in the dark for 16 h. It was washed two times with PBS and then examined under a microscope to identify blue-colored SA-β-gal-positive cells. The tissues’ reactive oxygen species and apoptosis were determined by fluorescent staining following a previously described method [56,57]. A detailed methodology is provided in Supplementary Method S2.

### 4.9. Statistical Analysis

SPSS (version 29.0, Chicago, IL, USA) software was used to determine one-way ANOVA after Tukey’s post hoc analysis to evaluate the statistical difference between the groups. For the pairwise statistical analysis, a Student’s *t*-test was performed.

## 5. Conclusions

This comparative consumption study concludes BWA’s substantially higher protective effect over CoQ_10_ against the HC+Gal-induced detrimental effect in zebrafish. BWA showed an affirmative impact on oxidative variables and restored sulfhydryl content hindered by the consumption of HC+Gal. BWA, owing to its antioxidant nature, impact on ROS generation, apoptosis, and cellular senescence paves a way to protect the kidney, testis, and ovary from the deleterious effects of HC+Gal. The findings suggest that incorporating BWA into the diet helps to improve antioxidant status and protects the kidney and reproductive organs from oxidative stress caused by excessive cholesterol and galactose intake.

## Figures and Tables

**Figure 1 pharmaceuticals-18-00017-f001:**
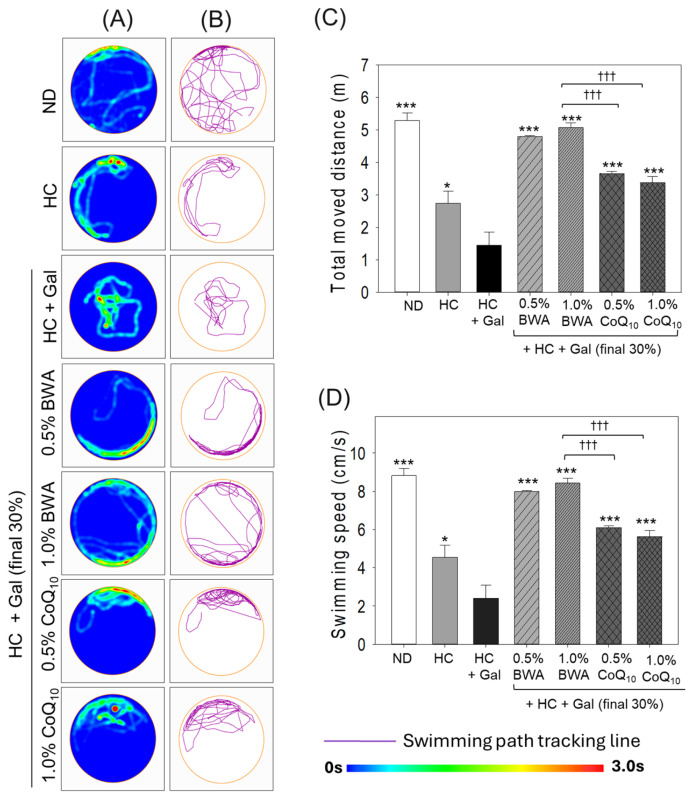
Overview of zebrafish motion captured from a top-down perspective (bird’s eye view) at 24 weeks of dietary intervention with high cholesterol and D-galactose. (**A**) Heat map illustrating zebrafish swimming pattern. (**B**) Visual representation of swimming trajectory. (**C**) and (**D**) Quantification of mean swimming distance and speed (cm/s), respectively. Seven zebrafish (n = 7) from each group were examined for 1 min using software (Any-Maze version 7.0, Kim & friends, Seoul, Republic of Korea) to analyze average swimming distance and speed. ND, HC, Gal, BWA, and CoQ_10_ are abbreviations for normal diet, high cholesterol, galactose, beeswax alcohol, and coenzyme Q_10_, respectively. * (*p* < 0.05) and *** (*p* < 0.001) depict statistical significance concerning the HC+Gal group, while ^†††^ (*p* < 0.001) highlights statistical difference compared to the BWA 1.0% group.

**Figure 2 pharmaceuticals-18-00017-f002:**
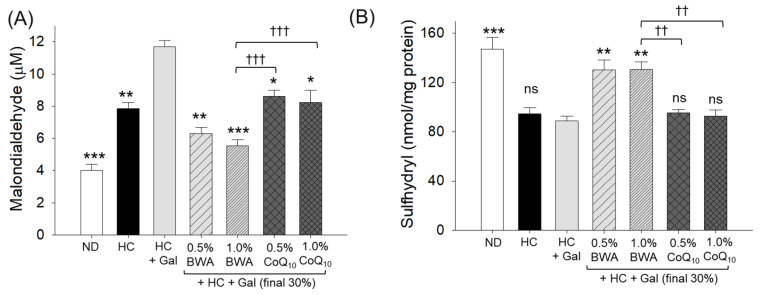
Oxidative variables in zebrafish plasma from different groups after 24 weeks of consumption of the specific diets along with high cholesterol and high galactose. (**A**) Quantification of the lipid peroxidation [assessed by thiobarbituric acid reactive substance (TBARS) assay using malondialdehyde (MDA) as the standard]. (**B**) Sulfhydryl content quantification. * (*p* < 0.05), ** (*p* < 0.01), and *** (*p* < 0.001) depict statistical significance concerning the HC+Gal group, while ^††^ (*p* < 0.01) and ^†††^ (*p* < 0.001) highlight statistical differences compared to the BWA 1.0% group; ns represents a non-significant difference between groups.

**Figure 3 pharmaceuticals-18-00017-f003:**
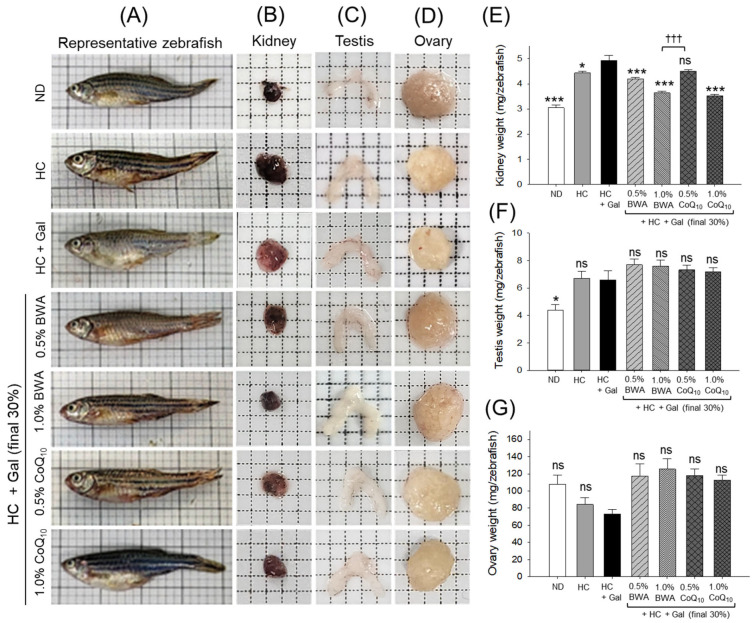
Images of different organs obtained from the zebrafish following 24 weeks of consumption of the designated diets under the influence of high-cholesterol and high-galactose diets. Representative images of (**A**) whole zebrafish, (**B**) kidney, (**C**) testis, and (**D**) ovary. The average weight of the (**E**) kidney, (**F**) testis, and (**G**) ovary. * (*p* < 0.05) and *** (*p* < 0.001) depict statistical significance concerning the HC+Gal group, while ^†††^ (*p* < 0.001) highlights statistical difference compared to the BWA 1.0% group; ns represents a non-significant difference between groups.

**Figure 4 pharmaceuticals-18-00017-f004:**
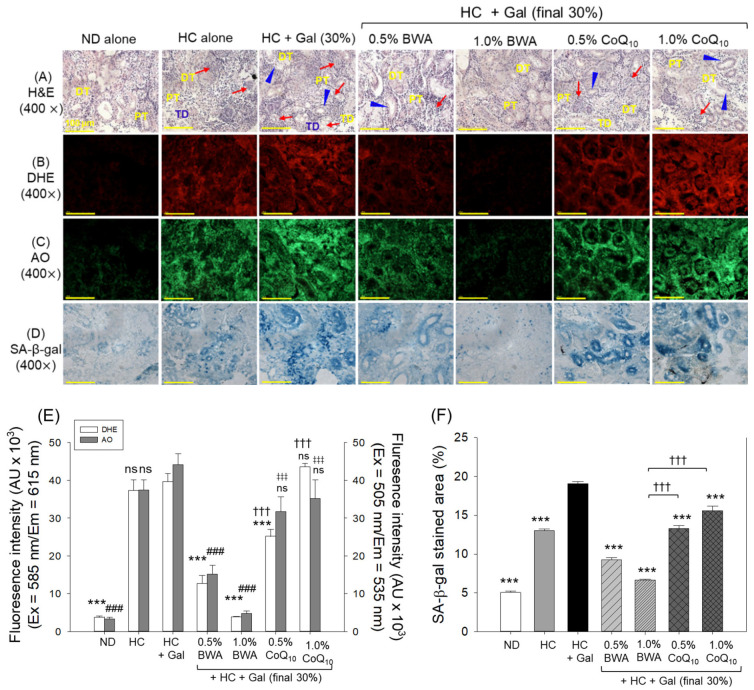
Histological analysis of zebrafish kidneys following 24 weeks of consumption of designated diets under the influence of high cholesterol and galactose supplementation. (**A**) Hematoxylin and eosin (H&E) staining; DT and PT depicting the distal and proximal tubules, respectively. The red arrow highlights the luminal debris, while the blue arrowhead indicates elevated tubular lumen. Scale bar = 100 μm. (**B**) Dihydroethidium (DHE) and (**C**) acridine orange (AO), fluorescent staining. (**D**) Senescent-associated β-galactosidase (SA-β-gal) staining. (**E**) Image J-based DHE and AO fluorescent staining quantification. (**F**) SA-β-gal-stained area. *** (*p* < 0.001) depicts statistical significance (for DHE and SA-β-gal-stained area), while ^###^ (*p* < 0.001) depicts statistical significance (for AO fluorescent intensity) concerning the HC+Gal group. ^†††^ (*p* < 0.001) highlights statistical difference (for DHE and SA-β-gal-stained area), and ^‡‡‡^ highlights statistical difference (for AO fluorescent intensity) compared to the BWA 1.0% group; ns represents a non-significant difference between groups.

**Figure 5 pharmaceuticals-18-00017-f005:**
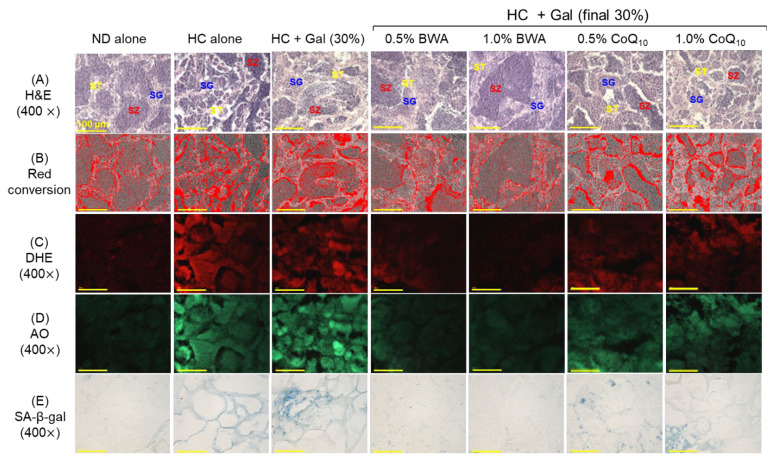
Histological analysis of testes obtained from zebrafish after 24 weeks of consumption of designated diets under the influence of high cholesterol and galactose supplementation. (**A**) Hematoxylin and eosin (H&E) staining; SG, ST, and SZ depicting the spermatogonia, spermatocytes, and spermatozoa, respectively. Scale bar = 100 μm. (**B**) Red conversion of the images of H&E staining to enhance visibility. The red conversion was performed using Image J-software (version 1.53, https://imagej.net/ij; accessed on 16 June 2023) at the white color threshold value (0-120). (**C**) Dihydroethidium (DHE) and (**D**) acridine orange (AO), fluorescent staining. (**E**) Senescent-associated β-galactosidase (SA-β-gal) staining. Quantification of (**F**) interstitial space between seminiferous tubules and (**G**) spermatozoa, (**H**) DHE and AO fluorescent intensity, and (**I**) SA-β-gal-stained area. * (*p* < 0.05), ** (*p* < 0.01), and *** (*p* < 0.001) depict statistical significance concerning the HC+Gal group (for interstitial space, spermatozoa area, DHE, and SA-β-gal area), while ^#^ (*p* < 0.05), ^##^ (*p* < 0.001) and ^###^ (*p* < 0.001) depicts statistical significance (for AO fluorescent intensity) concerning the HC+Gal group. ^†††^ (*p* < 0.001) highlight statistical differences (for interstitial space, spermatozoa area, DHE, and SA-β-gal area), and ^‡‡^ (*p* < 0.01) highlights statistical differences (for AO fluorescent intensity) compared with the BWA 1.0% group; ns represents a non-significant difference between groups.

**Figure 6 pharmaceuticals-18-00017-f006:**
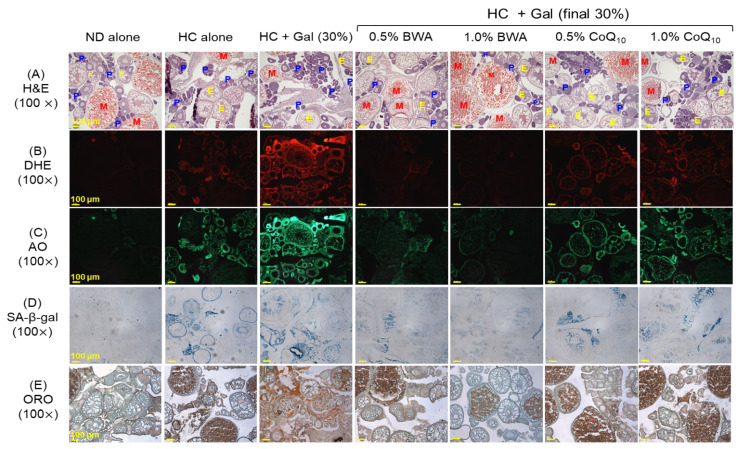
Histological analysis of the zebrafish ovaries after 24 weeks of consumption of designated diets under the presence of high cholesterol and galactose supplementation. (**A**) Hematoxylin and eosin (H&E) staining; P, E, and M depicting the pre-, early, and mature oocytes, respectively. Scale bar = 100 μm. (**B**) Dihydroethidium (DHE) and (**C**) acridine orange (AO), fluorescent staining. (**D**) Senescent-associated β-galactosidase (SA-β-gal) staining. (**E**) Oil red O (ORO) staining. Quantification of (**F**) pre-vitellogenic and (**G**) early and mature vitellogenic oocytes; five distinct sections for each group were examined for the quantitative estimation of pre-, early, and mature oocyte counts. (**H**) Image J-based DHE and AO fluorescent intensity quantification. (**I**) Quantification of SA-β-gal and ORO-stained areas. * (*p* < 0.05), ** (*p* < 0.01), and *** (*p* < 0.001) depict statistical significance (for pre-, early vitellogenic oocytes, DHE, and SA-β-gal area), while ^#^ (*p* < 0.05), ^##^ (*p* < 0.001) and ^###^ (*p* < 0.001) depict statistical significance (for AO fluorescent intensity and ORO-stained area) concerning the HC+Gal group. ^†^ (*p* < 0.05) and ^†††^ (*p* < 0.001) highlight statistical difference (for pre-vitellogenic oocytes, DHE, and SA-β-gal area) and ^‡^ (*p* < 0.05) and ^‡‡‡^ (*p* < 0.001) highlight statistical difference (for AO fluorescent intensity and ORO-stained area) compared to the BWA 1.0% group; ns represents a non-significant difference between groups.

**Figure 7 pharmaceuticals-18-00017-f007:**
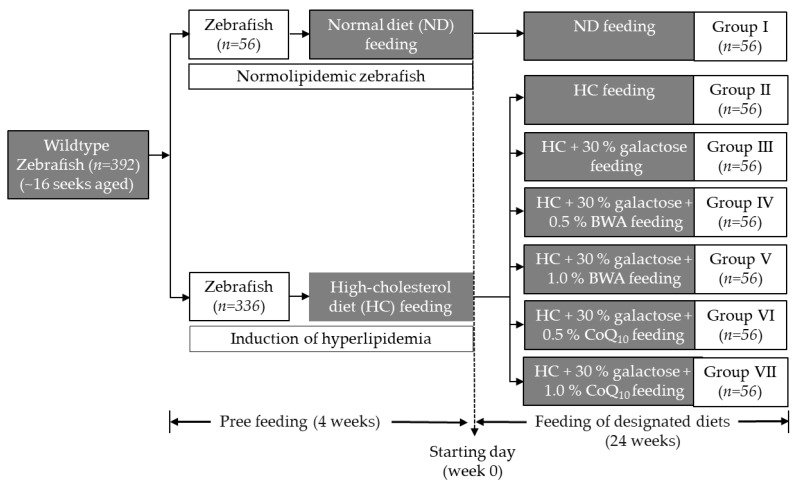
The experimental design involves zebrafish feeding on different diets: ND (normal diet), HC (high-cholesterol diet), BWA (beeswax alcohol), and CoQ_10_ (coenzyme Q_10_).

## Data Availability

The data used to support the findings of this study are available from the corresponding author upon reasonable request.

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
