# Peer review of "Beeswax Alcohol (BWA, Raydel^®^) Improved Blood Oxidative Variables and Ameliorated Severe Damage of Zebrafish Kidneys, Testes, and Ovaries Impaired by 24-Week Consumption of a High-Cholesterol and High-Galactose Diet: A Comparative Analysis with Coenzyme Q_10"

_pharmaceuticals, 2024, doi:10.3390/ph18010017_

Round 1
Reviewer 1 Report
Comments and Suggestions for Authors
This study takes an interesting and innovative approach by comparing the effects of beeswax alcohol (BWA) and coenzyme Q10 (CoQ10), showing that BWA clearly outperforms CoQ10 in protecting organs from damage caused by a diet high in cholesterol and galactose. The use of zebrafish as a research model is a good choice – these small fish share significant metabolic and genetic similarities with humans, making the results promising.
The findings clearly demonstrate that BWA has strong antioxidant and anti-aging properties, thoroughly examined through various methods, such as histological staining, analysis of oxidative stress markers, and cell apoptosis. All of this is backed by solid statistical data, which strengthens the conclusions. Notably, BWA performs better than CoQ10 in reducing oxidative stress markers and preventing damage to the kidneys, testes, and ovaries.
A major strength of the study is the clarity of its presentation – the results are well-organized, and the comparisons between BWA and CoQ10 are consistent and easy to follow. This makes it clear that BWA is not just another supplement but a potentially valuable compound with real significance in preventive medicine. When it comes to practical applications, the conclusions are likely to interest both researchers and companies working in the field of functional nutrition.
To sum up, this is a solid piece of work that brings something new to the field of health protection. The results are well thought out and backed by reliable data, showing that BWA’s can be a natural agent for preventing aging and organ damage.
Author Response
This study takes an interesting and innovative approach by comparing the effects of beeswax alcohol (BWA) and coenzyme Q10 (CoQ10), showing that BWA clearly outperforms CoQ10 in protecting organs from damage caused by a diet high in cholesterol and galactose. The use of zebrafish as a research model is a good choice – these small fish share significant metabolic and genetic similarities with humans, making the results promising. The findings clearly demonstrate that BWA has strong antioxidant and anti-aging properties, thoroughly examined through various methods, such as histological staining, analysis of oxidative stress markers, and cell apoptosis. All of this is backed by solid statistical data, which strengthens the conclusions. Notably, BWA performs better than CoQ10 in reducing oxidative stress markers and preventing damage to the kidneys, testes, and ovaries.
Response: Thank you for your understanding and appreciation.
A major strength of the study is the clarity of its presentation – the results are well-organized, and the comparisons between BWA and CoQ10 are consistent and easy to follow. This makes it clear that BWA is not just another supplement but a potentially valuable compound with real significance in preventive medicine. When it comes to practical applications, the conclusions are likely to interest both researchers and companies working in the field of functional nutrition. To sum up, this is a solid piece of work that brings something new to the field of health protection. The results are well thought out and backed by reliable data, showing that BWA’s can be a natural agent for preventing aging and organ damage.
Response: Thank you for your understanding and appreciation.
Reviewer 2 Report
Comments and Suggestions for Authors
Overall, the paper is excellently written and rich in extremely well-presented results, and I congratulate the authors on their excellent research.
Some comments/suggestions:
Line 419 (subtitle 4.2.) “husbandary” should be replaced with “husbandry”,
Lines 439-445: This part could be rewritten: each group should first be defined (write its name and the number of individuals it contains), and then state what it was fed with. If possible, provide a schematic diagram of the experimental design.
DISCUSSION: Perhaps you could say something more about the possibility of using Beeswax alcohol (BWA) in human and veterinary medicine, because high cholesterol consumption and high sugar consumption, as well as accompanying adverse events, are common in humans and domestic animals.
Author Response
Thank you for your insightful comments. Following the reviewer’s suggestion, we made point-to-point response and reflected on revision.
Please find attached doc as our response.

Reviewer 3 Report
Comments and Suggestions for Authors
- Keywords: Include “beeswax alcohol” on the list.
- Methodology: Add a section discussing the swimming behavior assay.
- A graphical abstract is highly recommended.
- The English language is generally fine but requires minor revisions.
Author Response

(The authors gave the same response as above.)
